# 3D shear wave velocity imaging of the subsurface structure of granite rocks in the arid climate of Pan de Azúcar, Chile, revealed by Bayesian inversion of HVSR curves

Rahmantara Trichandi[1,2,*], Klaus Bauer[1], Trond Ryberg[1], Benjamin Heit[1], Jaime Araya Vargas[3], Friedhelm von Blanckenburg[1,4], Charlotte M. Krawczyk[1,2]

[1]GFZ German Research Centre for Geosciences, Telegrafenberg, 14473 Potsdam, Germany
[2]Technische Universität Berlin, Ernst-Reuter-Platz 1, 10587 Berlin, Germany
[3]Departamento de Geología, Universidad de Atacama, Copiapó, Chile
[4]Institute of Geological Sciences, Freie Universität Berlin, Berlin, Germany

*Correspondence to*: Rahmantara Trichandi (chandi@gfz-potsdam.de)

**Abstract.** Seismic methods are emerging as efficient tools for imaging the subsurface to investigate the weathering zone. The structure of the weathering zone can be identified by differing shear wave velocities as various weathering processes will alter the properties of rocks. Currently, 3D subsurface modelling of the weathering zone is gaining increasing importance as their results allow the identification of the weathering imprint in the subsurface not only from top to bottom but also in three dimensions. We investigated the 3D weathering structure of monzogranite bedrock near the Pan de Azúcar National Park (Atacama Desert, Northern Chile), where the weathering is weak due to the arid climate condition. We set up an array measurement that records seismic ambient noise, which we used to extract the horizontal-to-vertical spectral ratio (HVSR) curves. The curves were then used to invert for 1D shear wave velocity (Vs) models, which we then used to compile a pseudo-3D model of the subsurface structure in our study area. To invert the 1D Vs model, we apply a trans-dimensional hierarchical Bayesian inversion scheme, allowing us to invert the HVSR curve with minimum prior information. The resulting 3D model allowed us to image the granite gradient from the surface down to ca. 50 meters depth and confirmed the presence of dikes of mafic composition intruding the granite. We identified three main zones of fractured granite, altered granite, and the granite bedrock in addition to the mafic dikes with relatively higher Vs. The fractured granite layer was identified with Vs of 1.4 km/s at 30 – 40 meter depth, while the granite bedrock was delineated with Vs of 2.5 km/s and a depth range between 10 and 50 meters depth. We compared the resulting subsurface structure to other sites in the Chilean coastal cordillera located in various climatic conditions and found that the weathering depth and structure at a given location depends on a complex interaction between surface processes such as precipitation rate, tectonic uplift and fracturing, and erosion. Moreover, these local geological features such as intrusion of mafic dikes can create significant spatial variations to the weathering structure and therefore emphasize the importance of 3D imaging of the weathering structure. The imaged structure of the subsurface in Pan de Azúcar provides the unique opportunity to image the heterogeneities of a rock preconditioned for weathering, but one that has never experienced extensive weathering given the absence of precipitation.

## 1. Introduction

Weathering can modify the mineralogy of rocks, resulting in petrophysical changes that can be detected with geophysical methods. Numerous geophysical imaging techniques have been successfully applied to characterize the weathering structure in granitic rocks, including the use of electrical resistivity tomography (Olona et al., 2010; Holbrook et al., 2014), Ground Penetrating Radar (GPR) (Dal Bo et al., 2019) and seismic tomography (Trichandi et al., 2022; Befus et al., 2011; Olona et al., 2010; Handoyo et al., 2022). These non-invasive techniques allow to estimate the weathering structure over wide extents and/or in areas of deep weathering, where probing using traditional techniques (e.g., soil pits) is logistically expensive or can be even unfeasible.

Seismic body wave tomography is one of the most frequently used methods in weathering zone studies (Befus et al., 2011; Flinchum et al., 2018; Olona et al., 2010; Trichandi et al., 2022). The seismic signals generated by sources such as hammer, weight drop, or vibroseis are recorded by an array of seismic receivers. Seismic P-wave velocity can be inferred by analysis and inversion of body waves' travel time. However, data acquisition of such body wave tomography experiments can be extensive as it requires the appropriate source and receiver array setup. It also restricts data acquisition in remote areas where heavy equipment transportation can be challenging. Moreover, active seismic experiments are often not permitted in conservation areas such as national parks, where the impact of the seismic source can disturb the surrounding ecosystems.

The Horizontal-to-Vertical Spectral Ratio (HVSR) technique is a passive seismic method that does not rely on active seismic sources. It has emerged as a viable alternative to active seismic methods for effectively imaging the sub-surface weathering zone. The HVSR method had initially been applied for site characterization in seismic hazard assessment (Piña-Flores et al., 2020; Mahajan et al., 2012) and imaging of sedimentary basins (Cipta et al., 2018; Pastén et al., 2016; Koesuma et al., 2017; Pilz et al., 2010). More recently, the HVSR method been utilized at smaller scales for investigating bedrock (Maghami et al., 2021; Moon et al., 2019; Nelson & McBride, 2019; Trichandi et al., 2023). This approach has gained popularity due to its simplicity, rapid data acquisition, and ease of placing measurement points. However, it is important to note that the inverse problem associated with the HVSR curve is highly non-linear and necessitates prior knowledge of the subsurface structure (Moon et al., 2019; Nelson & McBride, 2019; Pilz et al., 2010).

Utilizing a trans-dimensional and hierarchical Bayesian Markov chain Monte Carlo (McMC) approach solves the prior information requirement challenge in HVSR curve inversion. This approach is beneficial as it only requires minimal prior information about the sub-surface structure during the inversion process (Ryberg and Haberland, 2019; Trichandi et al., 2022). Moreover, the trans-dimensional approach allows for changes in dimensionality throughout the inversion, eliminating the need for prior constraints on determining the number of layers (Cipta et al., 2018; Bodin, 2010; Bodin et al., 2012). Consequently, this procedure reduces uncertainty levels and provides a robust, data-driven Vs (shear wave velocity) model that can be readily interpreted.

In this paper, we explored the shallow (< 50 m) subsurface structure of granite bedrock in the arid climate near the Pan de Azúcar National Park, which is one of the test sites of the "EarthShape (Earth Suraface Shaping by Biota)" project that cover

the Chilean climate gradient from dry to humid (Oeser et al., 2018; Oeser & von Blanckenburg, 2020). In this study, we apply the HVSR method combined with the Bayesian inversion approach to produce a 3D subsurface image of the weathering structure around an existing borehole (-26.302717° N, -70.457350° W). Then, we compare the results with previous studies conducted at the other EarthShape test sites to study further the effects of different climate and geological processes on the deep weathering of granite.

## 2. Study site and geological setting

The study was conducted at the outskirt of the Pan de Azúcar National Park, located in the Atacama Desert, approximately 200 km north of Copiapó, Chile (Figure 1A). The area is located in the arid part of the Coastal Cordillera featuring very low precipitation of ~10 mm/year and a mean annual temperature of 18.3 °C (Karger et al., 2017; Oeser et al., 2018). With such low precipitation, the vegetation in this area is minimal and is mainly sustained by the coastal fog coming in from the Pacific Ocean (Lehnert et al., 2018). A Ground Penetrating Radar (GPR) study in the Pan de Azúcar National Park managed to image the subsurface down to 2 meters depth, revealing an average soil thickness of 0.20 m (Dal Bo et al., 2019). Information from greater depth was not obtained due to the frequency limitation (Dal Bo et al., 2019).

The study area is in the Triassic Cerros del Vetado pluton (Figure 1B), comprising monzogranites and syenogranites (Godoy and Lara, 1998). Reported U-Pb ages for this pluton are in the range of 205-250 Ma (Berg and Baumann, 1985; Maksaev et al., 2014; Jara et al., 2021). Along its eastern margin, the Cerros del Vetado pluton is intruded by the Late Jurassic Las Ánimas pluton (Figure 1B), which has reported ages in the range of 150-160 Ma (Dallmeyer et al., 1996; Godoy and Lara, 1998; Jara et al., 2021). The Cerros del Vetado pluton is also intruded by a dense network of dikes with varying directions and compositions (Berg and Baumann, 1985; Acevedo, 2022), from which a swarm of NE-SW-oriented and dark-colored dikes of mafic (gabbroic to dioritic) composition is clearly observed in the field and crosses the study area (see dark-colored lines in Figure 1B and C). Dallmeyer et al. (1996) report a $^{40}Ar/^{39}Ar$ age of 129.2 ± 0.5 Ma for one of these dikes that outcrops 8 km south-west of the study area. In 2019, a drilling campaign was conducted to investigate the weathering structure by extracting drill cores to perform core analyses. The location of the borehole is shown in Figure 1D.

According to investigations of samples from the borehole, the bedrock at Pan de Azúcar is characterized by a high $SiO_2$ (Ø 74 wt%), $Na_2O$, and $K_2O$ content (> 7 wt%) and remarkably uniform major element concentrations. It comprises the major minerals of quartz, plagioclase, microcline, and biotite. The grain size distribution is relatively homogenous, with an average grain size around 0.5 cm. The granite has been hydrothermally overprinted to variable degrees. This alteration is strongest in the vicinity of large fractures. Televiewer data from the borehole revealed that fractures dip angles show a continuous range of dip angles between 20 and 80°. Fractures of shallower dip alternate with steeper dipping fractures throughout the entire core. Fractures are partly cemented with calcite, clay minerals, and Fe-oxides. Bedrock porosity is, with an average of around 1%, generally low, whereas the hydrothermally overprinted core material has a bulk porosity of exceeding 2.5%. Weathering mass balances using the refractory element niobium indicated zero weathering, as expected in arid climate. Mean soil

denudation rates measured at the drill site with in situ cosmogenic [10]Be measured in quartz from surface soil is $7.1 \pm 0.5$ t km-2 yr[-1] (n=3).

## 3. Data Acquisition

We conducted a passive seismic data acquisition in Pan de Azúcar in August 2022 in a 150 x 150 m area covering the borehole location (Figure 1C) with elevation varying between 710 – 750 meters. The investigated area extends from gently sloping bedrock in the southeast into a colluvial valley fill consisting of sandy gravel in the northwest. The distance between measurement points was set to be 10 meters apart, resulting in 225 measurement points. The ambient noise data was acquired by the subsequent deployment of 15 3-components (3-C) geophones with 4.5 Hz Eigenfrequency (PE-6/B by SENSOR Nederland). Since we expected a shallow weathering structure, the 4.5 Hz Eigenfrequency is deemed to be sufficient to cover the desired depth range. The geophones were planted into the ground and were connected to a CUBE data logger (DATA-CUBE³ Type 1 with internal GPS – by DiGOS), which records the seismic vibrations detected by the geophones. We simultaneously deployed 15 geophones along a northwest – southeast direction line and recorded the ambient seismic noise for 1 hour for each measurement point. To cover the whole area with 225 measurement points, we required four days of fieldwork, including the array setup.

## 4. Methodology

To image the subsurface using the Horizontal-to-Vertical Spectral Ratio (HVSR) method, we first need to process the recorded 3-component noise data and extract the HVSR curve. When extracting the curves, we followed the statistical approach described by Cox et. al. (2021). To guarantee sufficient data quality, we excluded the curves that did not satisfy the Site EffectS using AMbient Excitations (SESAME) Guideline for site characterization (SESAME, 2004). Then, we inverted each curve using a trans-dimensional, hierarchical Bayesian inversion approach to get a 1D Vs profile from each measurement point. We then used the multiple 1D profiles to produce a pseudo-3D Vs model of the subsurface by smoothing and interpolation. A geochemical analysis of the borehole's core samples was also performed to provide a weathering indicator to the modelled Vs data.

### 4.1. HVSR curve extraction

The HVSR technique is a method to retrieve information of the subsurface seismic properties using a single station measurement on the surface. This was initially developed for site effect investigations (Nakamura, 1989) based on the frequency spectrum ratio of the horizontal and vertical components of a seismic recording at the same location. To extract the HVSR curves, we used the geometric mean of the frequency spectrum between the horizontal and vertical components of the seismic recording which can be described in the following equation:

$$HVSR(\omega) = \frac{\sqrt{H_{EW}(\omega) \times H_{NS}(\omega)}}{V(\omega)} \tag{1}$$

Where $H_{EW}(\omega)$ and $H_{NS}(\omega)$ are the frequency spectrums of the horizontal east-west and north-south seismic components, and $V(\omega)$ is the frequency spectrum of the vertical seismic components. Sites with relatively hard and strong rock theoretically will not amplify the energy within a certain frequency band. Therefore, the ratio between the horizontal and vertical seismic components will roughly be equal. On the other hand, sites with weak and softer rocks will significantly amplify the frequency spectrum energy at a certain frequency band and results in a higher HVSR value (Xu and Wang, 2021). Therefore, the frequency where the HVSR curves peak can be used to determine the interface between bedrock and its overlying rocks.

Prior to the HVSR curve extraction process, we applied instrumental correction and a 1.0 to 150 Hz bandpass filter to the three components' raw data (vertical, east-west, and north-south). We used the 60 minutes of ambient noise for each point, as shown by the example in Figure 2A – C. The time-series data were then transformed into a frequency domain by applying a straightforward Fourier transform (Figure 2D – F). To improve the quality and stability of our HVSR curve, we followed the statistical approach of HVSR curve extraction described by Cox et. al. (2021). We divided the time series data into 60 seconds time windows and rejected time windows that exceeded a statistical threshold of peak frequency and variance. This windowing approach will reject time window which results in high standard deviation of the peak frequency. Therefore, high frequency events which could contaminate the ambient noise records such as the high frequency and amplitude event recorded around 2,800 second mark in Figure 2A – C will be excluded from the dataset as they will result in high standard deviation compared to the other time window. Ultimately, we obtained the final HVSR curve and the corresponding standard deviation of the HVSR curve shown in Figure 2G.

After extractions of the HVSR curve from all the observation points, we checked for the HVSR curve's consistency. Color plots of the HVSR curve in Figure 3A reveal mainly two frequency ranges with the peak frequency: below 30 Hz and around 100 Hz. Since the study is more focused on the boundary between the bedrock and its overlaying weathering structure, the higher frequency data will later be discarded as the HVSR curve peak around 100 Hz is often associated with the boundary between soil and saprolite (Nelson and McBride, 2019). The HVSR curve with high uncertainties were also discarded. An average of all the HVSR curves is shown in Figure 3B, where it is possible to observe various frequency peaks in the frequency range lower than 10 Hz, which could be related to the varying bedrock depth.

## 4.2. HVSR curve inversion

The substantial non-uniqueness inherent in the HVSR curve modelling introduces a highly non-linear inverse problem. Various approaches have been employed to address this challenge, including the implementation of a least-square approach (Arai and Tokimatsu, 2004), fixing the number of layers (Fäh et al., 2003; Parolai et al., 2005; Hobiger et al., 2013; Wathelet et al., 2004), and fixing the bedrock velocity (Parolai et al., 2005). These approaches tried to simplify the inverse problem by incorporating prior information about the true model into the inversion process. For instance, it is often difficult to determine

the expected number of layers without preliminary subsurface investigations in the context of investigating weathering zones.

As for fixing the bedrock velocity, although previous research on similar bedrock formations can provide some prior knowledge about the velocity, it is common for bedrock velocities to vary between different sites (Trichandi et al., 2023, 2022). Furthermore, reliably fixing the bedrock velocity would necessitate accurate bedrock P- and S-wave velocity information. For the inversion workflow, we followed the trans-dimensional hierarchical Bayesian Markov Chain Monte Carlo (MCMC) inversion scheme as described by Bodin (2010), which had also been applied to the HVSR curve inversion (Cipta et al., 2018;

Trichandi et al., 2023). The main advantage of using this inversion scheme is that we require only minimum prior information about the real model (Ryberg and Haberland, 2019; Trichandi et al., 2022). With this inversion routine, we do not need to fix the bedrock velocity as it will be determined during the inversion process. Furthermore, with the trans-dimensional approach, we do not have to define a fixed number of layers for our model, as the number of layers will be determined during the inversion process based on the data noise level.

For the parameterization of the inversion, we defined a wide range of values for each parameter. The Vs value was determined to be between 0.1 and 5.0 km/s to accommodate the possible existence of very loose material, such as soil with low velocity or even intruding solid rocks layer with very high velocity. The advantage of using a relatively large model space is also to allow the inversion scheme to explore the whole model space and not be trapped in a local minimum. For the number of layers, we also enabled the inversion to explore models with a number of layers between 2 and 20 layers (including half space). As

for the input data, we limit the observed HVSR curve to be between 1 and 20 Hz.

Forward calculation of the HVSR curve also requires an explicit definition of the number of layers. We accommodated the inversion to explore both simple and complex layered models by performing a trans-dimensional inversion approach. While the conventional approach often subdivides weathering layers into three or four layers (soil, saprolite, fractured bedrock, bedrock), we allow the inversion to explore a simple two-layered model. This simple layered model accommodates situations

where only fractured bedrock and bedrock layers are present, particularly in areas with a relatively low degree of weathering. Furthermore, this simplified model is beneficial for incorporating data with high uncertainty. Conversely, we also allowed a maximum of 20 layers to account for the possibility of highly intricate sub-surface layering, acknowledging the potential complexity that may be present in certain areas.

The subsequent parameters to be determined are Vp and density. Previous studies have demonstrated that these parameters

exhibit relatively low sensitivity to HVSR curve modeling and can be considered unknown nuisances during the inversion process (Cipta et al., 2018). However, for completeness, we also established uniform ranges for Vp, ranging from 0.3 to 8.0 km/s, and for density, ranging from 1.5 to 3.0 g/cm3. It is also important to note that we did not invert the density value in our inversion and used the inverted Vp value to approximate the density using the equation provided by Brocher (2005).

We performed the inversion for each data point on 24 parallel Markov chains with 100,000 iterations per chain. Each chain

starts by drawing a random initial model ($m_0$) from the given prior information. The initial model is then used to model a HVSR curve ($dcal_0$) using the GEOPSY forward modelling package (Wathelet et al., 2020). The forward calculation tool employed a model representing the sub-surface as a series of homogeneous layers extending into a half-space. We used a 1D

Voronoi cell approach of constant velocity layer when constructing the layered model. Then for every iteration, one of the following perturbations was performed:

- Perturb the data noise (σ) value according to the proposal distribution.
- Perturb the Voronoi cell location of a random cell according to the proposal distribution. Changes in the Voronoi cell location will affect the layer's thickness of the input model.
- Perturb the Vs value of a random cell according to the proposal distribution.
- According to the proposal distribution, perturb a random cell's Vp/Vs value and calculate Vp accordingly for the
input model.
- Delete one random cell.
- Add one random cell based on the prior information provided by the chain.

The perturbation process generates a new model ($m_1$), from which we calculate the corresponding HVSR curve ($dcal_1$). Subsequently, we determine the acceptance probability for transitioning from the initial model ($m_0$) to the new model ($m_1$)
based on the observed data ($dobs$), the calculated HVSR curve of the initial model ($dcal_0$), and the calculated HVSR curve of the new model ($dcal_1$). We randomly decide whether to accept the transition from $m_0$ to $m_1$ using this acceptance probability. If the transition is accepted, $m_1$ replaces $m_0$, and we proceed to the next iteration. However, if the transition is not accepted, we retain $m_0$ and move on to the next iteration. This process is repeated until the specified number of iterations is reached. The final model for each station is obtained by averaging all the accepted models from the 24 Markov chains that were executed.

By averaging all the accepted models, the final model now contained information of the extensively sampled model space and can bring information that is not contained if we only use a single best model approach (Trichandi et al., 2023). Additionally, we also include the model uncertainty by calculating the standard deviations of all the accepted models. We then used the uncertainty to limit the model depth.

### 4.3. Chemical depletion factor

In addition to the seismic properties, we also measured the Chemical Depletion Factor (CDF) from the borehole's core samples. CDF can be used to quantify the fraction of mass loss by weathering relative to the bedrock (Riebe et al., 2003; Krone et al., 2021) . We calculated the CDF on core samples from the Pan de Azúcar drill core from concentrations of the insoluble element Nb in the parent rock and compared it the Nb concentration in the weathered bedrock. The CDF is the calculated using the following equation:

$$CDF = 1 - \frac{[Nb]_{bedrock}}{[Nb]_{weathered\ rock}} \qquad (2)$$

Where $[Nb]_{bedrock}$ is the Nb concentration from bedrock sample, and $[Nb]_{weathered\ rock}$ is the Nb concentration from the target weathered rock sample. From Equation 2, a positive CDF value means that the target weathered rock sample went through a depletion while a negative CDF value means the rock went through an enrichment of the Nb concentration. For the CDF calculation in Pan de Azúcar, we compared the Nb concentration from the fractured and hydrothermally altered cores

with the relatively unweathered granite. An example of a strongly fractured and hydrothermally altered core can be found in Figure 4A and Figure 4C, while Figure 4B shows a relatively unweathered granite.

## 5. Results

We divided our results into two main parts: (1) results from the HVSR curve and the subsequent (2) pseudo-3D Vs model. Even before performing the HVSR curve inversion, the HVSR curve already brings an insight into the weathering zone depth as the peak frequency ($f_0$) from the HVSR curve is often linked to the top of the bedrock depth (Nelson and McBride, 2019; Nagamani et al., 2020; Stannard et al., 2019). Using $f_0$, we can approximate the depth of the top of the bedrock using the following equation:

$$D = \frac{1}{4f_0}V_s \tag{3}$$

Where D is the depth to the top of the bedrock, and Vs is the average S-wave velocity of the overlying layer. This equation shows that higher $f_0$ is related to a shallower top of the bedrock depth, while low frequency $f_0$ indicates a deeper top of the bedrock depth. With this approximation, we can estimate the fresh bedrock depth across the study area. However, this equation highly depends on the assumption of correct Vs and therefore can only be used as an approximation.

### 5.1. HVSR curve frequency profile

After extracting the HVSR curves, we can determine the $f_0$ of each HVSR curve and produce the peak frequency and apparent fresh bedrock depth map (Figure 5A and B). The peak frequency map in Figure 5A can be created by plotting each measurement point's peak frequency. The high frequency in Figure 5A is related to a shallower interface between the regolith and bedrock, while low frequency means a deeper interface as shown by the depth approximation equation in Equation 1. Figure 5B shows the apparent bedrock depth obtained using equation 1, assuming a Vs of 1.5 km/s for overlaying layer. This value is reasonable as we expect the bedrock to be overlain by a fractured bedrock which can have a Vs up to 1.5 km/s (Trichandi et al., 2022, 2023).

From the peak frequency and apparent fresh bedrock depth map, we can already see an interesting feature in our study area. The most distinctive feature is the two southwest–northeast stripes of high $f_0$– related to the shallow interface to the bedrock layer. However, certain precautions must be taken when interpreting this apparent fresh bedrock depth map, especially as the apparent fresh bedrock depth formula assumes a homogenous value of the layer velocity and a sharp Vs contrast between the layers. Nevertheless, the peak frequency and apparent bedrock depth map still serve as a good indicator of potential contrasting subsurface structure. In the study area, the peak frequency is likely related to the interface between fractured bedrock and bedrock unit, as we do not find any indication of a saprolite unit in the study area.

## 5.2. Results from borehole location

We present an the 1D S-wave velocity profile result of a data point located 1 meter away from the borehole (Figure 6) to demonstrate the robustness of the Bayesian inversion routine and compare it to the calculated CDF values. The modelled HVSR curve matched well with the observed curve and is well within the modelled standard deviation (Figure 6A). Figure 6B shows how the inversion routine explored the Vs model space within the widely defined Vs corridor. In the end, the 1D Vs model was derived from the average of all the Vs models, as shown as the blue line in Figure 6B. The uncertainty of the model

is also presented as the dashed black line. Here we can see that at the borehole location, the Vs down to 28 – 30 meters depth are relatively constant, followed by a slight increase from 30 to 40 meters depth. Starting from a 40-meter depth, a slight increase in Vs value can be observed until 55 meters depth. Below this, the increase in Vs is not as steep, but we can still see an increase until around 40 meters deep, where we start losing the resolution due to the high uncertainty of the half-space models.

Since we utilize Voronoi cells in the model parameterization, the Bayesian inversion also produces an interface probability model function (Figure 6C). This function shows the probability of a likely interface between the layers based on the models produced from all the inversion chains. The interface probability offers four possible interfaces in our data around 15, 28, 48, and 60 meters in depth. We also present the calculated CDF in Figure 6D. The CDF values in the near surface (< 20 m) shows neither a trend of enrichment or depletion of the rock samples. Instead, we observe a relatively scattered CDF values around

zero. We also observe similar scattering trend around 50 – 60 m depth and 80 m depth.

## 5.3. S-wave velocity depth profile

We compiled the resulting 1D Vs model and interpolated them to create a volumetric cube of Vs value. A series of depth slices from 10 – 50 meters depth is presented in Figure 7, where we can see a similar NE – SW feature as in Figure 5. This feature can be observed from 10 meters depth and is getting stronger at 20 meters. At smaller depths, the velocity is relatively low (<

1.50 km/s) except around the previously mentioned NE-SW features, with the lowest velocity of around 1.00 km/s. These low-velocity features remain until 30 meters depth (Figure 7C) where the high velocity starts to take over the low-velocity values around the area. Then at 40- and 50-meters depth, the remaining low-velocity area is located northwest of the borehole.

We also plot a horizontal depth slice of 28-meter depth / 693 meters above sea level (masl) (Figure 8A) following the end of the altered granite layer interpreted in the borehole core interpretation (Figure 6E), and to further investigate the interesting

NE-SW features, we also create a vertical slice of our data perpendicular to the NE-SW features and going through the borehole location (Figure 8B). The vertical slice shows varying Vs between 1.0 – 3.0 km/s. Horizontally, we do not see so many variations except in the western part of the profile, where we can see a relatively lower velocity layer right before the slope. Vertically, we can divide the profile into three main Vs layers. The first section is the layer with Vs < 1.40 km/s which covers down to 15 meters in depth. This section is relatively parallel to the surface topography. Then, we have the section with Vs

between 1.40 and 2.40 km/s which covers a depth between 15 and 30 meters and shows more horizontal variations compared

to its overlaying layer. Finally, the section with Vs > 2.40 km/s starts around 30 meters in depth near the dashed line reference in Figure 8B.

In addition to the vertical slice across the borehole, we present a vertical section north of the borehole location (Figure 9). Similar to the profile in Figure 8B, the Vs value in this profile also ranges between 1.0 and 3.0 km/s. However, instead of the relatively parallel homogenous horizontal variations in Figure 8B, in this profile, we can see an interesting near-vertical structure of high-velocity ca. 10 meters below the surface (Figure 9B).

## 6.    Discussion

Using the HVSR method combined with the hierarchical Bayesian McMC inversion approach, we managed to image the 3D Vs velocity structure in the arid study site of Pan de Azúcar. We interpret and discuss possible weathering zone structures in the study area using the velocity model. In summary, four main structures in the study area were found: granite bedrock, fractured granite, basaltic intrusion, and hydrothermally altered granite. In addition to the identified structures, we also discuss different controlling processes that could explain the dynamics and formation of the weathering structure in the study area, including the effects of topography. Finally, we compared the structure in Pan de Azúcar to the other EarthShape sites in different climate conditions.

### 6.1.  Granite bedrock

Various Vs values had been used to delineate the upper boundary of physically and chemically unweathered intact granite bedrock formation. Previous studies conducted in similar granitic environments have suggested that a Vs value of 2.0 km/s is an appropriate threshold between the granite bedrock and the overlaying layer (Liu et al., 2022; Handoyo et al., 2022). Within the context of the Earthshape project, another geophysical investigation conducted in a site with a Mediterranean climate (La Campana National Park, see location in Figure 1A) has indicated that we can utilize a Vs value of 2.3 km/s to identify the upper limit of the bedrock layer (Trichandi et al., 2023). In a site with a semi-arid climate (Santa Gracia National Park, see location in Figure 1A), another study revealed that a Vs value of 2.5 km/s is appropriate for defining the upper boundary of the bedrock interface (Trichandi et al., 2022). We can attribute the selection of different Vs values to determine the bedrock's upper limit to either variation in lithological composition or the impact of distinct climatic conditions. Nevertheless, given the similarity in climate, we opted to employ a Vs value of 2.5 km/s to establish the upper boundary of the bedrock in our study area.

Based on the Vs value across the area in Figure 7, we can see that the 2.5 km/s Vs value started at different depth ranges. In Figure 7A and B, we can see that even at shallow depths of 10 and 20 meters, Vs is already at 2.5 km/s. However, we believe that the NE-SE trending high-Vs zones in the middle of the study area are related to dikes of mafic composition, which we will discuss in section 6.3. Therefore, we believe that the physically intact bedrock starts from 30 meters deep at the shallowest, as shown by the high Vs value in the south-western part of the area shown in Figure 7C. This high Vs value is then becoming

more prominent in the 40- and 50-meter depth slice (Figure 7D and E), where we can see that the high Vs value is spreading to the north-east direction. Based on this 3-dimensional observation, we can already see that instead of a uniform bedrock depth, as shown in Figure 8B, the top of the bedrock depth is more variable in the different directions. An intact bedrock depth between 30 to 50-meter depth agrees with other previous studies that found intact bedrock depth in an arid and semi-arid area to be between 30 to 70 meters depth (Stierman and Healy, 1984; Vázquez et al., 2016; Trichandi et al., 2022).

## 6.2. Fractured granite

The presence of fractures in the bedrock leads to a systematic decrease in the Vs values. Previous studies have shown that Vs values between 1.3 – 1.4 km/s can be used to identify the top of the fractured bedrock layer (Trichandi et al., 2022, 2023). Based on the horizontal depth slice of the Vs model (Figure 7), we can already find Vs > 1.3 km/s even at 10 meters depth (Figure 7A) even when we exclude the NE-SW high-velocity features.

The low velocity areas are of Vs < 1.3 km/s can be explained by several lithology, including saprolite, colluvial deposits, or highly fractured granite. Saprolite layer from granitic rocks typically will show a much lower Vs value (< 1.0 km /s) (Trichandi et al., 2022, 2023). Data provided by the borehole also shows no indication of a saprolite layer. For colluvial deposits, field observations did observe the existence of colluvial valley between the two hills, but the extent of this valley fills was not further investigated. Since colluvial valley will usually shows lower Vs values (< 1.0 km/s) (Handoyo et al., 2022), we opted to attribute this unit as granite rocks with significantly higher amount of fractures within. However, it is also possible that the thickness of this colluvial valley is well below the vertical resolution of our HVSR method and therefore we do not observe its low Vs signature.

The vertical extent of the identified fractured bedrock layer seems to vary significantly across the area: while on the west of the borehole Vs value larger than 2 km/s are reached at depths shallower than 20 m (blueish zones in Figure 7C), in the northern part of the area Vs value lower than 1.3 km/s is observed down to 40 m depth (brownish colors in Figure 7D). This variation across the study area showcases the importance of 3D mapping in weathering front study, as it is easy to conclude a relatively uniform thickness of the fractured granite layer if we only consider a 2D profile across the borehole (Figure 8B).

Several processes can explain the fracturing of bedrock in our study area. Firstly, we can hypothesize that the fractures were inherently formed during the cooling process of the plutonic rocks (Ellis and Blenkinsop, 2019). Should this be the case, fractured bedrock in our study area was likely to date back to the emplacement of the Cerros del Vetado, which took place ca. 250-205 Ma. This emplacement likely happened in a depth range of 4 – 7 km (as most geobarometry studies of plutonic rocks of the Coastal Cordillera and Precordillera of northern Chile, e.g., Dallmeyer et al., 1996; Dahlström et al., 2022). Another possible explanation for the formation of this fractured granite layer is the process of lithostatic decompression, occurring at shallow depths due to the subduction-induced uplift and erosion of the entire coastal region. Considering that current denudation rates from cosmogenic nuclides are only 7.1 t/m$^2$/yr which corresponds to about 2.7 m/million years the fracturing would have taken place within the past several million years.

## 6.3. Mafic dikes

The Vs model consistently shows two NE-SW oriented zones of relatively higher Vs visible from 10 to 50 m depth (Figure 7). This feature is observed on multiple station location and shows a systematic high Vs values. We interpret these features to be related related to the Early Cretaceous NE-SW mafic dikes, which crop out around the study area and intrude on the Triassic Cerros del Vetado pluton (Figure 1B, D, and E). This hypothesis can explain the higher Vs values as the gabbro and diorite values are higher than granite (e.g., Christensen, 1996). The quasi-linear distribution of outcrops of these dikes in the study

area (even when crossing small ridges, see Figure 1B) suggests that they have high-angle dips, consistent with the vertical shape of high Vs zones (Figure 9B).

Hypothetically, we can assume if the mafic dikes intrusion could be the features which create the fractured granite unit. However, it is unlikely (and difficult to prove) that the current fractured granite layer was already present when the dikes emplaced (130 Ma), and that the fractured granite layer has been there over such a long time. According to thermochronological

studies in the Coastal Cordillera of northern Chile (e.g., Juez-Larré et al., 2010; Rodríguez et al., 2018), there have been at least two significant exhumation events during the Cretaceous and the Eocene. If there was a shallow fractured granite layer during the emplacement of dikes ca. 130 Ma, such fractured granite layer likely was eroded after some of those exhumation events.

Based on the Vs model in Figure 9B, we create a conceptual lithology model in our cross-section where the dike feature is

prominent (Figure 9C). From the bottom, we have the granite bedrock which were intruded by two mafic dikes. Then from ca. 30 meters depth to the surface, we have an overlying altered and fractured granite. The mafic dikes in our deployment area also do not seem to intrude up to the surface as we did not find any exposed dike in the middle of the profile during the data acquisition. This is also supported by the high Vs value that does not reach to the surface. Another possible explanation as to why we did not find surface manifestation of the dike could be because of the colluvial fill which covers the middle area of

our survey area. Nevertheless, based on this result and findings, we demonstrated the capability of HVSR method to identify a higher Vs structure in the subsurface even with minimal prior information.

## 6.4. Altered granite

Chronologically, the Cerros del Vetado pluton emplacement took place 250-205 Ma and likely in a depth range of 4-7km (as most geobarometry studies of plutonic rocks of the Coastal Cordillera and Precordillera of northern Chile, e.g., Dallmeyer et

al., 1996; Dahlström et al., 2022). Then, the Las Ánimas pluton emplacement took place 160-150 Ma. This later emplacement might have induced hydrothermal alteration in the Cerros del Vetado pluton unit, as identified in the borehole. However, the emplacement of mafic dikes into the Cerros del Vetado pluton granites can also carry heat and possible fluids to enable hydrothermal alteration. The mafic dikes intruded the Cerros del Vetado pluton ca. 130 Ma (i.e., almost 100 Ma after Cerros del Vetado emplacement).

Cores from the borehole shown in Figure 4A and C shows an example of the identified altered granite unit in our borehole location. The altered granite exhibits a relatively reddish colour compared to a relatively intact granite bedrock shown in Figure 4B which indicates oxidation of iron minerals. These altered granite units can also be identified by the scattering of the CDF values shown in Figure 6D. The identified altered granite units also appears to weaken the granite rock as shown in Figure 4A and C. Weakening of the granite rock in the hydrothermally altered zone is likely due to the formation of secondary mineral

which can also triggers micro-fracturing even at depth (Hampl et al., 2023). These weakening of the granite rock will then negatively affect the Vs value.

    Since the hydrothermal alteration of the granite rock negatively the Vs value, one can also try to interpret the low Vs near the surface across the study area to a hydrothermally altered granite layer. While we lack the information to prove otherwise, we hypothesize that the altered granite in our study area can only be found in close proximity to the possible mafic dike structures.

These mafic dike structures provide the heat and fluids to alter the intruded granite rocks around them. However, we also cannot eliminate the possibility of an interconnected fracture network which allows heat and fluid to be transported and enable hydrothermal alteration further away from the mafic dike structures.

### 6.5. Controlling aspects of weathering structure

Based on the lithology identified in the study area, we summarised several significant controlling aspects that can affect the weathering structure. The first important aspect which could induce a particular weathering structure is the existence of fractures. Previous studies have shown fractures' importance in facilitating weathering processes (Trichandi et al., 2022; Brantley et al., 2017; Lodes et al., 2023). These fractures can either be formed due to weathering or tectonic processes. The Pan de Azúcar study site is located in a very dry area which limit the chemical weathering process triggered by precipitation.

The absence of chemical weathering is also supported by the CDF values shown in Figure 6D where we observe no clear trend of Nb element enrichment. This is also evident as there is no saprolite layer found in the borehole. Despite the absence of chemical weathering process, we still observe a reduced seismic velocity which can be attributed to fractured granite layer. Therefore, we hypothesize that the formation of the fractured granite layer in our study area to be formed by tectonic stress or lithostatic decompression instead of chemical weathering process.

The absence of water and vegetation cover in the Pan de Azúcar study sites presents a unique opportunity to study fractured bedrock that precedes weathering processes. The fractures formed in the bedrock, when connected, can act as a pathways for fluid or other weathering agents which triggers subsurface chemical weathering of the fractured bedrock. However, since our study site in Pan de Azúcar has no precipitation and vegetation cover, the observed fractured bedrock is likely to be isolated from other weathering processes. This observation is highly unlikely to be found in most locations on Earth with precipitation

or vegetation cover which enable physical, chemical, or biological weathering processes.

    In addition to the availability of fractures due to preconditioning of the bedrock, the existence of geological structures also plays a significant role in shaping the weathering structure in our study site. A Previous study in the Santa Gracia Earthshape

site shows how a possible fault could play an important role by allowing deeper weathering and providing fluid infiltration pathways (Trichandi et al., 2022). In Pan de Azúcar, the younger mafic dike intrusion could trigger fracturing, enabling

hydrothermal alteration of the surrounding rocks. These findings emphasize the importance of looking at weathering structure not only as a 1D top–down interaction but also as a 3D interplay with the surrounding geological features.

Finally, climate effects in forms of precipitation also highly affect the weathering structure, especially in the near-surface. While our study area shows exposed fractured granite to the surface, the almost absence of precipitation restricts the surface's weathering process, which in return does turn the fractured bedrock into saprolite. While it is yet to be seen whether fog covers

coming in from the Pacific Ocean can affect the weathering structure, our Vs model indicates the absence of saprolite as we do not encounter a very low Vs value (typically below 0.8 km/s for saprolite). The absence of the saprolite layer is also supported by the lack of a significant trend of enrichment or depletion in the near surface as shown by the CDF values in Figure 6D.

## 6.6. Weathering depth and climate gradient

We compare the findings from different seismic investigations of the weathering zone across the Chilean climate gradient in Figure 10. From Figure 10A, we can see a trend deepening of the average bedrock depth with increased precipitation. Our study using the HVSR method in Pan de Azucar reveals an average bedrock depth of around 30 meters depth. Similarly, in the semi-arid climate of Santa Gracia, evidence from a borehole (Krone et al., 2021) and seismic imaging revealed a weathering front that reaches down to around 30 meters in depth (Trichandi et al., 2022). If we observe only these two sites, we can

hypothesize that either increasing precipitation has no effect on the bedrock depth, or the increase of precipitation between the arid climate in Pan de Azucar to the semi-arid climate in Santa Gracia is not so significant that it does not affect the bedrock depth. However, seismic investigation using HVSR method in the Mediterranean climate of La Campana (Trichandi et al., 2023) show a relatively deeper average bedrock depth (~60 meters depth) compared to the arid and semi-arid sites. When we include this finding, we can see the trend of weathering front deepening with increased precipitation as we go from lower to

higher precipitation. This observation confirms that when we have a changing climate, increased in precipitation can deepen the weathering front. However, there is likely to be a certain threshold to be passed for the increase in precipitation to significantly affect the bedrock depth.

We also compare the effect of increased precipitation to the saprolite thickness (Figure 10B). In the arid climate of Pan de Azucar, there is saprolite layer due to the absence of the required weathering agents to transform (fractured) bedrock into

saprolite. Unlike its effect on the bedrock depth, precipitation changes between the arid and semi-arid climate of Santa Gracia seems to affect the saprolite thickness more significantly as we observe averagely 13 meters thick of saprolite across the seismic profile in Santa Gracia (Trichandi et al., 2022). This shows that even when both sites have similar depth extent of the fractured bedrock as the preconditioning to enable weathering processes, the climate condition in Pan de Azúcar does not satisfy the condition to trigger extensive weathering of the fractured bedrock to saprolite. The thickening of the saprolite layer

due to increase of precipitation is even more pronounced when we compare the saprolite thickness in Santa Gracia to the

mediterranean climate of La Campana where we observed a significantly thicker saprolite layer up to 55 meters thick (Trichandi et al., 2023).

This paper presents a unique case of bedrock which had been pre-conditioned (fractured) but have yet to be weathered. The depth of the bedrock does not deepen significantly when we go the semi-arid climate. However, the increase of precipitation

enables saprolite production as precipitation provides the required weathering agent to reach the subsurface through the available inherent fractures. Therefore, in addition to the depth of the weathering front, it is important to discuss the saprolite thickness when studying the effects of changing climate to the weathering zone structure as it correlates with the weathering intensity of the study area.

## 7.    Conclusion and outlook

The presented study showcases the importance of imaging the weathering zone, not only in 2D but also in a 3D point of view. Applying the HVSR method with the Bayesian inversion scheme provides a straightforward technique for 3D imaging of the weathering structure. The passive seismic nature of the HVSR method also enables data acquisition where usage of an active seismic source is restricted or even prohibited. The resulting 3D model also revealed dike features which we will not find if we only do a 2D imaging of the subsurface.

Based on the Vs model, we identified fractured and altered granite in Pan de Azúcar to reach 30 – 40 meters deep. There is no Vs signature of a saprolite layer that is indeed absent in Pan de Azúcar due to the arid climate as also shown by the CDF values of the borehole core. Mafic dike structures were identified in the study area which can be the source of hydrothermal alteration. Comparing the results of Pan de Azúcar (arid climate) with Santa Gracia (semi-arid) and La Campana (Mediterranean) EarthShape sites, we interpret that the depth and structure of weathering zone is a result of the complex interplay between fluid

infiltration, the presence of interconnected pathways allowing fluid to migrate to great depth, and the mineralogy of the bedrock before being exposed to surficial weathering. The imaged structure of the subsurface in Pan de Azúcar is unique in the sense that it shows a rock preconditioned for weathering, but one that has never experienced any extensive weathering given the absence of precipitation.

This study also further emphasizes the vital role of geophysical methods in imaging the weathering zone. This work shows

that the employed seismic method helped differentiate geological features with distinctive weathering conditions (e.g., mafic dikes and the granites) that can be further investigated using a method with high resolution (e.g., borehole). This study also shows the importance of imaging the weathering zone in 3D, as the structures of weathering zone can exhibit significant variations in the horizontal direction.

## 8. Data availability

The data that support the findings will be available in the GFZ Data Repository following an embargo to allow for doctoral publication of research findings.

## 9. Author contributions

RT, KB, and CK planned the campaign; RT and BH performed the measurements; RT, KB, and TR analyzed the data; RT wrote the manuscript draft; KB, TR, BH, JAV, FVB and CK reviewed and edited the manuscript.

## 10. Competing interest

The authors declare that they have no conflict of interest.

## 11. Acknowledgement

We acknowledge support from the German Science Foundation (DFG) priority research program SPP-1803 'EarthShape: Earth Surface Shaping by Biota' (Grant number KR 2073/5-1). Instruments for data acquisition were supported by the Geophysical Instrumental Pool Potsdam – GIPP (Grant number GIPP-201924). This work was also supported by EarthShape Coordination (EH 329/17-2, BL562/20-1). The authors are very thankful to K. Übernickel for her support in planning and performing the drilling campaign and downhole logging. We are also grateful to Brady Flinchum and one anonymous reviewer who gave invaluable comments and suggestions.

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

a

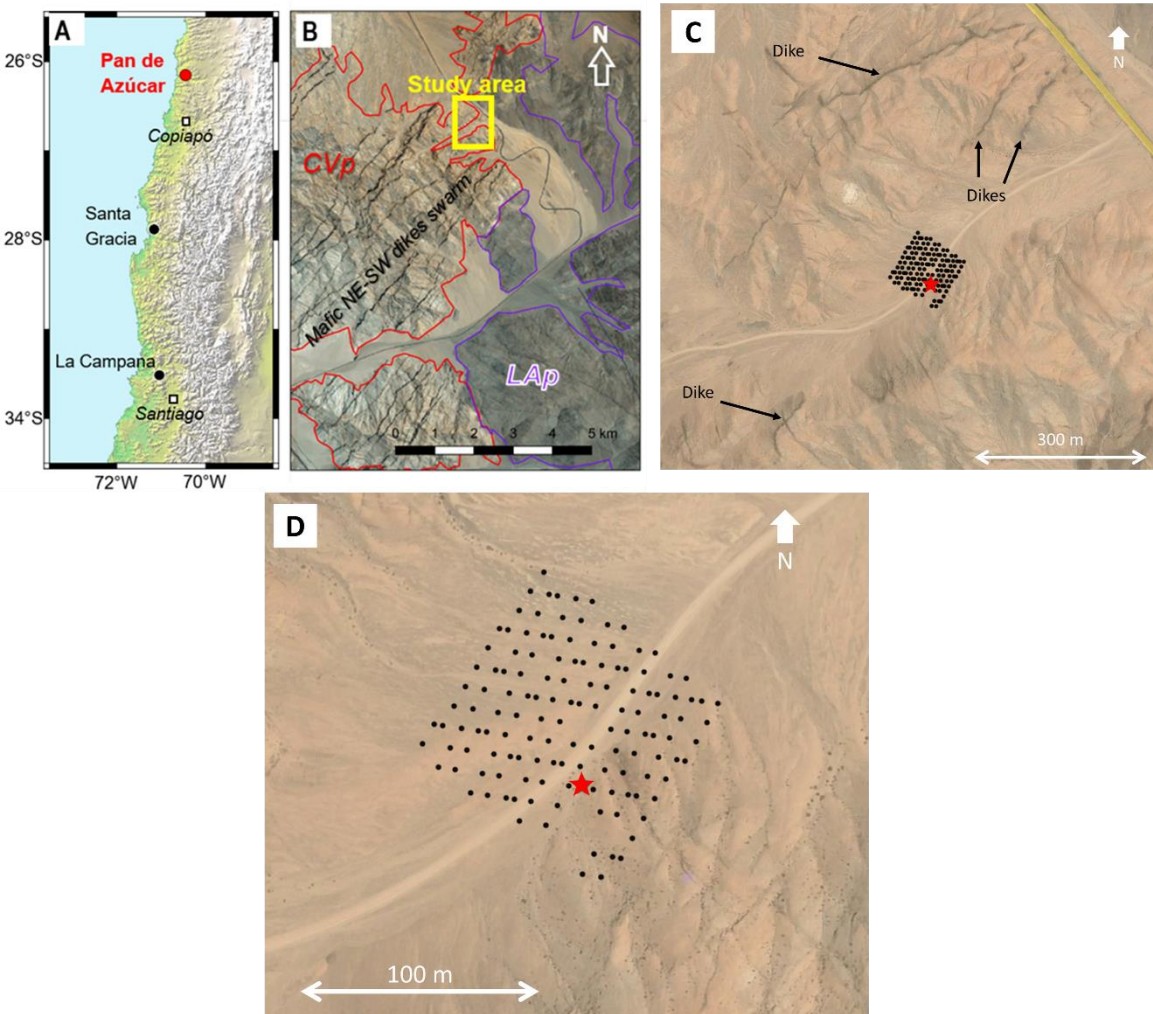

**Figure 1. Site information of the Chile Pan de Azúcar study site: (A) Study site location. (B) Local geology information of the study area (Sernageomin, 1982). Red and purple lines outline the Cerros del Vetado (CVp) and Las Ánimas (LAp) plutons, respectively (modified from Godoy & Lara, 1998). (C) Google Earth image of the study site (© Google Earth 2019). Some of the mafic dikes outcropping in the area are highlighted. (D) Seismic array. Black dots in (C) and (D) show the position of seismic sensors. The red star shows the location of the borehole (WGS 84 coordinates 26.30272° S, 70.45735° W).**

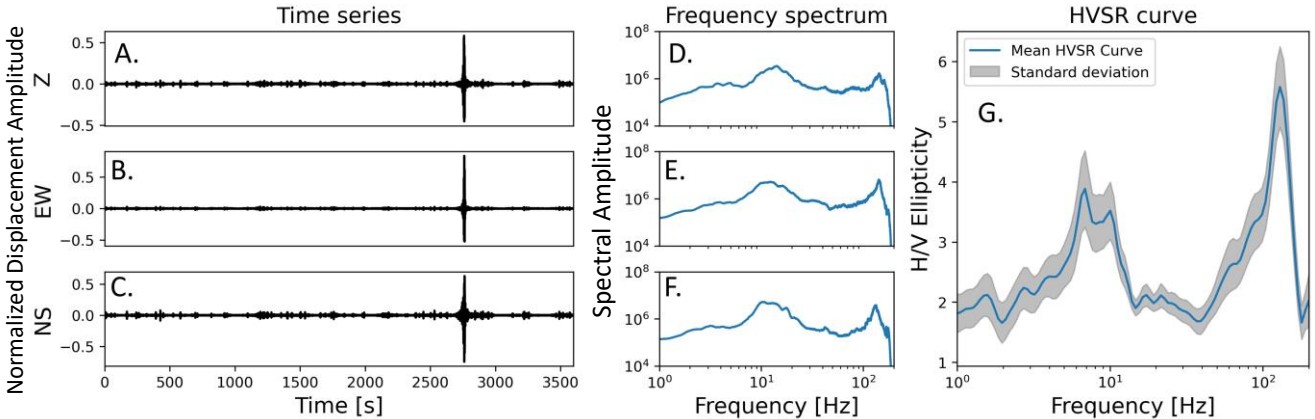

**Figure 2. Steps of HVSR curve extraction from a station at the borehole location (See Figure 1D). (A–C) 60 minutes 3 components time domain normalized seismic noise recordings, (D-F) the respective frequency power spectrum, and (G) the resulting HVSR curve extracted using the hvsrpy package (Vantassel, 2020).**

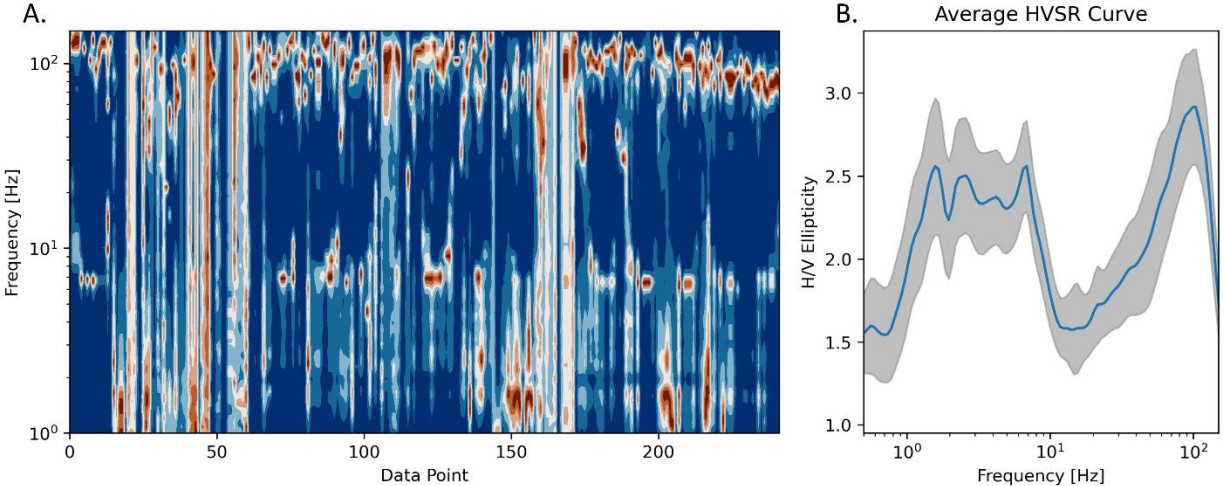

**Figure 3. Collection of the extracted HVSR curve in the survey area (A) and the mean HVSR curve (B).**

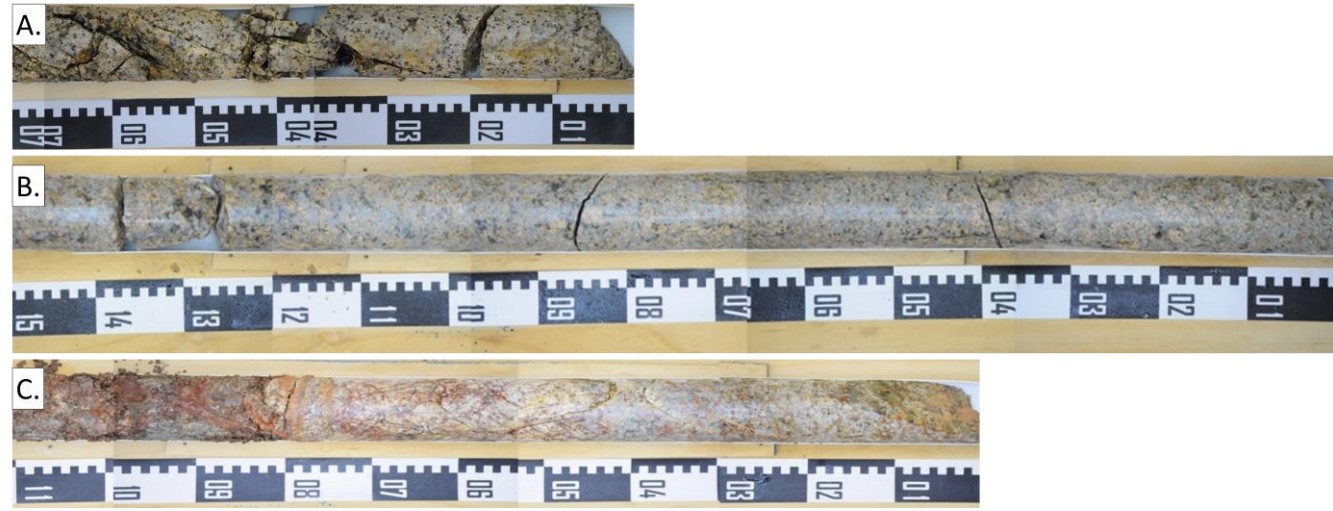

**Figure 4. Comparison of (A) strongly fractured and hydrothermally altered granite from 17 m core depth, (B) relatively unweathered granite from 72 m core depth, and (C) strongly hydrothermally altered granite from 86 m core depth.**

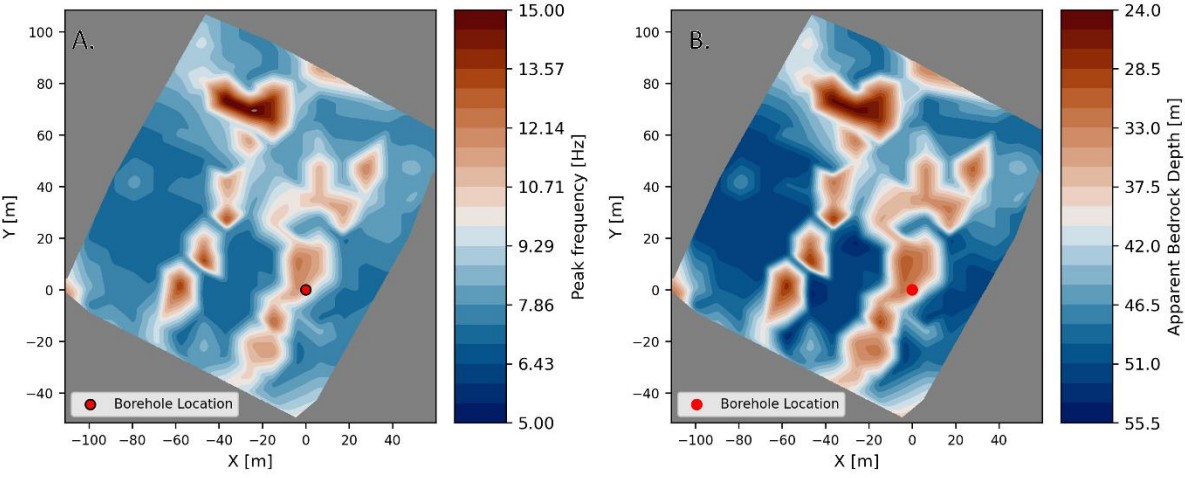


**Figure 5 Resulting (A) peak frequency and (B) apparent fresh bedrock depth map assuming Vs = 1.50 km/s. The red dot represents the borehole location. Higher frequency in (A) is related to a shallower bedrock, while inversely, lower frequency is connected to a deeper bedrock. Area with no data points are greyed out.**

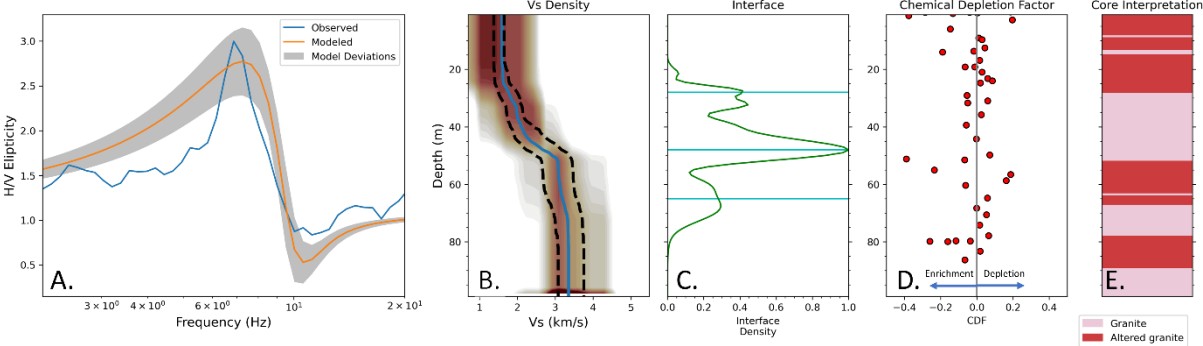


**Figure 6. Bayesian inversion result of the HVSR curve from the data points closest to the borehole location: (A) observed and modeled HVSR curve; (B) 1D Vs probability density from all the inversion chain and the mean Vs profile, blue line is the mean model, dashed black lines are the uncertainty of the models; (C) interface probability function; (D) CDF from borehole samples; and (E) interpretation from the borehole core.**


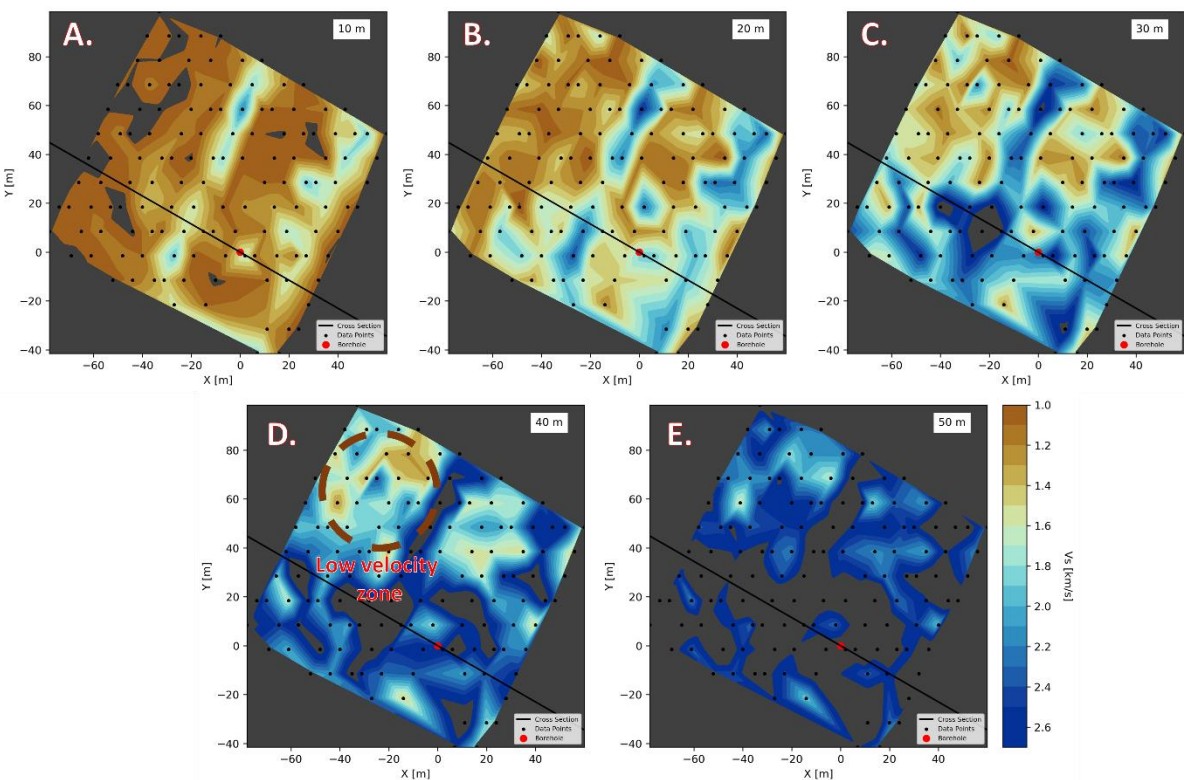

**Figure 7 Depth slice of the pseudo-3D Vs model produced from the inversion of HVSR curves. From (A) to (E) are depth slices with steps of 10 meters from the surface. The black dotsare the geophone location and the black line represents a NW-SE cross-section that goes through the borehole location of the red dot. Greyed out area are area which were not covered by the geophone location or has high uncertainty and thus were not resolved.**


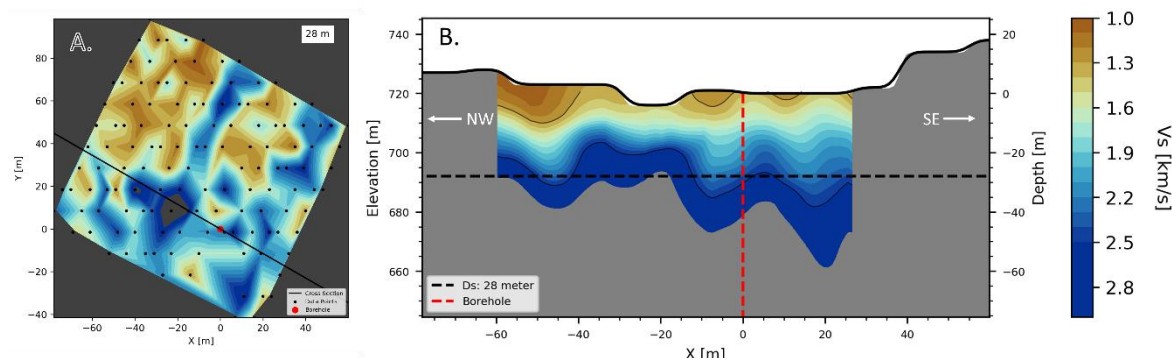

**Figure 8. Vs model of the study area: (A) horizontal depth slice at 28-meter depth, red dot is the borehole, black dots are the station location, and solid black line is the cross-section line.(B) vertical slice from the cross-section line in (A). The dashed black line in (B) represents the selected 28-meter depth line, the dashed red line represents the borehole location, solid black lines are the 1.30 and 2.50 km/s Vs contour lines. Depth axis in (B) is relative to borehole elevation (720 meter elevation). (A) and (B) used the same color bar presented on the right.**

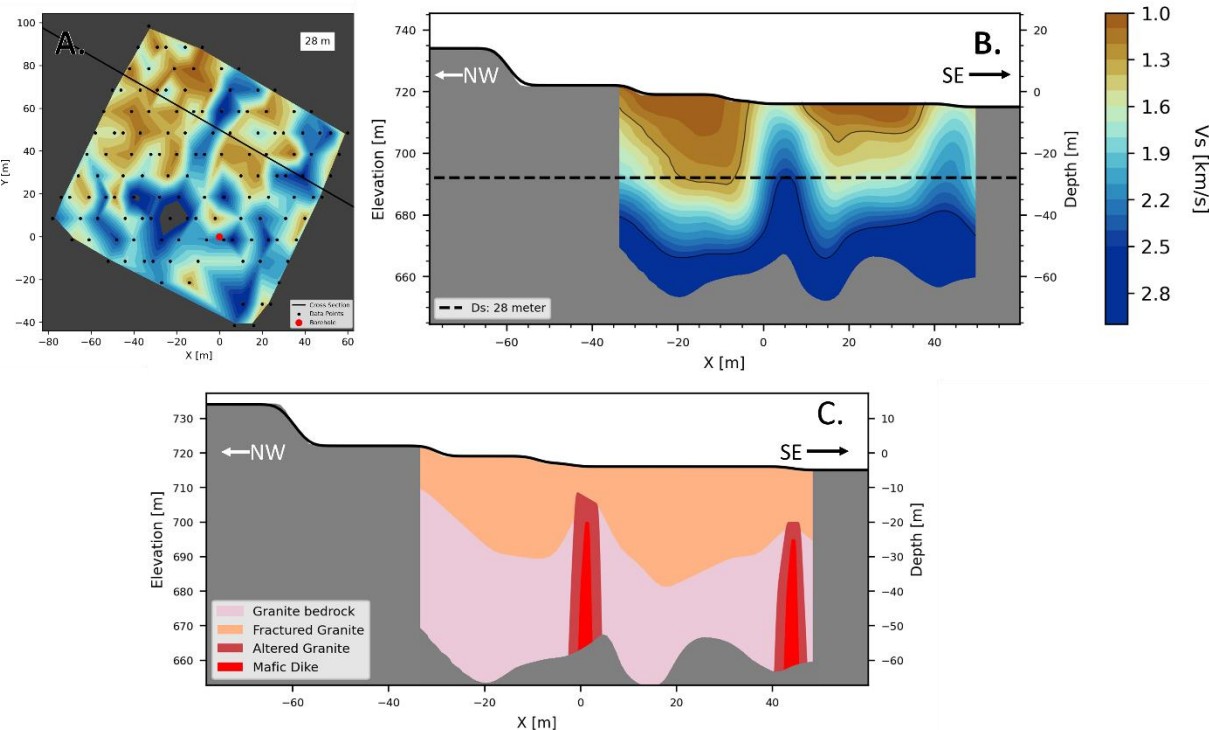

**Figure 9. Vs model of the study area: (A) horizontal depth slice at 28-meter depth, (B) vertical slice from the cross-section line in (A), and (C) conceptual model of the cross-section in (B). The red dot in (A) is the borehole location, and the inverted triangles are the geophone location. The dashed black line in (B) represents the selected 28-meter depth line.**

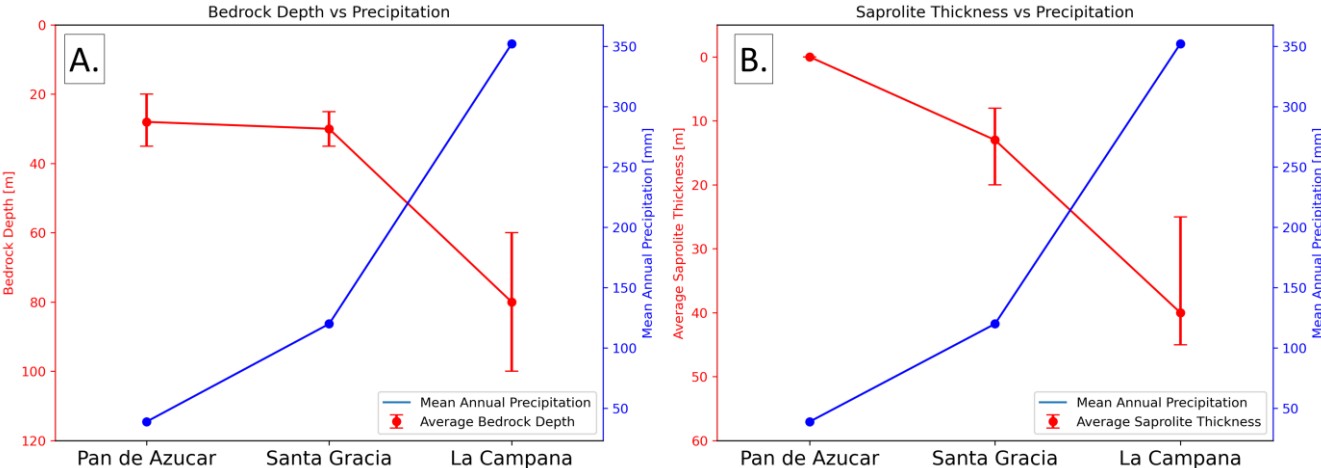

**Figure 10. Comparison of the precipitation to (A) bedrock depth and (B) saprolite thickness in different EarthShape site across the Chilean climate gradient. Red lines are (A) the average bedrock depth and (B) the saprolite thickness (Trichandi et al., 2022, 2023). Error bars shows the maximum and minimum value. Blue lines are the mean annual precipitation from the different sites (Werner et al., 2018)**