# Peer review of "3D shear wave velocity imaging of the subsurface structure of granite rocks in the arid climate of Pan de Azúcar, Chile, revealed by Bayesian inversion of HVSR curves"

_EGUsphere, 2023_

## Referee Comment (RC2)

**Review**

3D shear wave velocity imaging of the subsurface structure of granite rocks in the arid climate of Pan de Azúcar, Chile, revealed by Bayesian inversion of HVSR curves

By Rahmantara Trichandi et al.

**General Comments:**

This paper evaluates the 3D structure of monzogranite bedrock near the Pan de Azúcar National Park (Atacama Desert, Northern Chile), using Bayesian inversion of horizontal-to-vertical spectral ratios (HVSR). The authors analyzed a large number of HVSR curves, derived from ambient noises, recorded at over a dozen stations in the study region. The authors used their best HVSR curves to invert for 1D shear wave velocity models, which later used to generate 3D model beneath the study region. The results are summarized in **7. Conclusion and outlook:**

*Based on the Vs model, we identified fractured and altered granite in Pan de Azúcar to reach 30 – 40 meters deep. We found no Vs signature of a saprolite layer that is indeed absent in Pan de Azúcar due to the arid climate. Mafic dike structures were identified at depth.*

The technical quality of the paper is good although there are considerable grammar/writing mistakes that should be corrected before publishing. Authors have clearly analyzed and interpreted the results. Novelty of the research is also acceptable. However, I have some suggestions for authors to improve the quality of the manuscript before publishing it. Therefore, I recommend accepting this article with major revisions. My suggestions are as below:

The authors assembled a large data set and rigorously processed HVSR curves to generate 3D shear wave velocity model. However, it is somewhat disappointing that the presented results do not include resolution test (e.g., checkerboard test. See Wang et al., 2021) to estimate the horizontal and vertical resolution of the model. I would suggest adding that into the results section which will improve the quality of the manuscript.

It would be great to compare shear wave velocity profile in the borehole with the 1D profiles extracted at nearby stations. If any information available, please add that comparison into the manuscript.

**Specific comments:**

1. Line 24: Add "shear wave velocity (Vs)".

2. Line 44-45: Not clear what authors wanted to report here.

3. Line 50-53: Not clear. Re-write the sentence.

4. Line 89-90: Use subscript (e.g., $Na_2O$)

5. Line 104-106: Remove sentences start with " Initially…" and "However…"

6. Line 203: Remove "quickly"

7. Line 233: Define "masl"

8. Line 116: Define "SESAME"
9. Line 238: Please add velocity contour lines into figure 7b to show 3 layers discussed in the text.

10. Line 307: Replace "also ask" with "assume". I noted similar types of wording at many places in the manuscript (e.g., sentence start with "We found…" in line 377) and I would recommend fixing those.

11. Line 293: What is "βpo"

Figures:
Fig 1: Caption- Replace "Red dots" with "Black dots".

Fig 2: Please add Y-axis labels for figures A-F. Where is this receiver located? Add receiver location into figure 1. Are the records in velocity or displacement or raw data?

Fig 4: Add shading (e.g., gray color) to the area where you have less data coverage (white region in your plots) or lower resolution (see example figures in Wang et al., 2021).

Fig 5: Add data point location into figure 1. If Vs profile beneath the borehole is available, pleas add that into figure B for comparison.

Fig 6: Add shading as I mentioned above for Fig 4. Please add color scales corresponding to each slice. If you use same color scale for all figures, then, put only one-color scale bottom of the figure. Fig A-C needs x-axis labels. Remove words top on each figure (e.g., Depth Slice DS: XX) and add depth value upper right corner of each figure as "10 m or 20 m etc." (see example figures in Wang et al., 2021). Keep black solid line (cross section) within the area where you have the best resolution.

Fig 7 & 8: For Fig A; similar comment as Fig 6.

For Fig B; Zoom into the region where you have the best resolution. It seems like X axis is not correct. Remove text above the figures B and C. Mark 'NW' and 'SE' corner of the profile (see example figures in Wang et al., 2021). Remove depth axis in Fig B and C.

Reference suggested:

Wang, W., Nyblade, A., Mount, G., Moon, S., Chen, P., Accardo, N., et al. (2021). 3D seismic anatomy of a watershed reveals climate-topography coupling that drives water flowpaths and bedrock weathering. Journal of Geophysical Research: Earth Surface, 126, e2021JF006281. https://doi. org/10.1029/2021JF006281

---

## Author Response (AR1)

**Response Letter**

**Reviewer 1 (Brady Flinchum)**

We thanked the first reviewer for the extensive review which helps in improving our manuscript. We provided the following response to the comments:

**General Comments:**

1. ***Comments:*** *" ….Was there saprolite observed in the core shown in Figure 5?.... , I think there needs to be some more evidence supporting the idea that there is no saprolite at the site"*
   **Response:** We have now presented core pictures from the borehole as well as the Chemical Depletion Factor analysis to support the arguments that there are no saprolite in the study area. (Figure 4 and 6, Chapter 6.2)

2. ***Comments:*** *" From a geophysical standpoint, you can only determine if there are fractures, which is questionable. However, after reading through this, I think that I pieced together this argument: the authors believe these are tectonically or inherited fractures because this area is arid (but the data are not shown or given), if there was water present, then the velocities would be much lower than 0.8 m/s and there would be saprolite at the site. "*
   **Response:** We thank the reviewer for this comment. We include more information and discussion about the arid climate of the study area so that the though process as to why we interpret the fractures to be inherent or came from tectonic processes instead of weathering processes can be followed more easily. (Figure 10, Chapter 6.6)

3. ***Comments:*** *" The second part of this argument references another two sites (but not shown here) that weathering is deeper elsewhere. I think these are valid points, but they need to be articulated a bit better in this paper so that non-geophysicists, someone from regolith/critical zone science, will recognize the significance of the findings."*
   **Response:** As suggested, we include the findings from the other two sites to support the comparison of the weathering front which support the argument and extend the discussion of the different sites comparison. (Figure 10)

4. ***Comments:*** *"How important is setting the bedrock velocity in the HSVR inversion? For example, if you select a bedrock velocity that's too fast, what effect does that have on the inverted results? The authors could address this by giving an example or something, "Selecting bedrock velocities is critical because if we overestimate the velocity, we get a slightly shallower first layer". Or is this point irrelevant because the authors use an MCMS type of inversion and sample a full range of velocities?"*
   **Response:** While other inversion approached required reliable prior information such as the bedrock velocity or the number of layers, using the hierarchical Bayesian Markov Chain Monte Carlo (MCMC) inversion scheme allows us to reliable model the Vs based on the HVSR curves with little to no prior information as the inversion scheme sample a full range of velocities given. This is highly advantageous, especially as we do not have sonic log data available. (Chapter 4.2)

**Line Specific Comments:**

1. **Comments:** *L150 " From the standpoint of this paper, I don't know what the authors did without going back and reading Bodin (2010). Can the authors summarize the paper in 1-2 sentences?."*
   **Response:** We made the necessary addition to the paragraph. (Line 163 – 169)

2. **Comments:** *L150 "Why is the ratio between the Horizontal and Vertical important? The authors never actually give a single-sentence explanation for how HSVR works. It would be nice for readers more interested in the weathering story, as opposed to the geophysical inversion if that makes sense."*
   **Response:** We add a brief introduction about the HVSR method as suggested. (Line 122 - 134)

3. **Comments:** *L155 "If you know there is a weathered granite why have more than 4: soil/saprolite/fractured bedrock/bedrock?"*
   **Response:** We put high number of layer to accommodate smooth layer. This is now explained in more detail. (Line 176 - 183)

4. **Comments:** *L159 " So even though you let it vary to 20 layers the inversion results zeroed in on 3-4 layers? That's quite remarkable if all the uncertainties above are correct. I would suggest highlighting and emphasizing this point."*
   **Response**: Thank you for the comments. While it is true and remarkable that our models do converge to a 3 – 4 layers model, we do not use a single model (e.g., model with the lowest error) as the final model, but instead average all the accepted models from all the chain and iterations.  We do this so that the final model represents the huge model space that had been sample. We also believe that an average model presents a more realistic physical representation of the smooth subsurface instead of the blocky best model approach. (Line 176 – 183 and 210 - 213)

5. **Comments:** *L169 "What relationship did you use? Is it empirical is it linear? Density = m\*Vp ?? Or is it exponential, density = a\*Vp^x?"*
   **Response:**  We used the Brocher equation (Brocher, 2005) used exponential equation to approximate density based on Vp. (Line 188)

6. **Comments:** *L194 " In other words, in one to two sentences can you sum up those references? I also this would be a good place to put the first equation."*
   **Response:** Thank you for the suggestion, we have now added explanation which relates the HVSR curve and bedrock depth.  (Line 228 – 238)

7. **Comments:** *Figure 2 "How does the peak around 2800 s affect your HSVR inversion? Is that an earthquake? Is that an active source? I assume that's responsible for the peak around 100 Hz in panel G. What would have happened if you didn't have that spike. For example, in Figure 4, are the spikes of linear features a result of where the ambient energy is higher frequencies? ……. Can you show a sounding that doesn't have a huge spike that still has the 100 Hz peak present?  In other words can you make figure two so that it shows what woul appear to be*

*noise and still have the 100 Hz energy? This really makes me think at this stage in the manuscript (maybe you address it later) that high bedrock is a result of some external source creating higher frequencies."*

**Response:** Thank you for the question. It is correct that the existence of such high-amplitude, high frequency event such as the waveform around 2800 s could affect the HVSR curve. However, in our HVSR curve extraction process, we applied an approach which subdivides the data into multiple time windows. For example, the data shown in Figure 2 is a 3600 s records which we subdivided into 60 time windows. Then, we extract the HVSR curve for the individual time window and discarded the time windows that can make the standard deviations of the 60 HVSR curves higher. With this approach, we automatically discarded the time windows with high-amplitude and high-frequency. Therefore, we believe that the peak around 100 Hz in panel G does not related any high-frequency and high-amplitude event, but instead related more to very thin cover of soil/sand or colluvial deposits. We present a figure below similar to Figure 2 but without any high-frequency and high-amplitude event in the data which still show a peak around 100 Hz.

[Figure]

(Line 139 - 145)

8. **Comments:** *L250 " How do the authors know this is hydrothermally altered? Is this coming from the borehole logs?"*
   **Response:** Correct, this is coming from borehole core analysis. We have now presented the core geochemical analysis and an explanation for it. (Figure 4, Figure 6A, Chapter 4.2)

9. **Comments:** *L315 "Does the HVSR confirm a fast velocity at depth? …..Is it possible that the interpreted dikes are just lateral heterogeneity in the weathering profile, something like Tor coming to the surface? Does the HVSR confirm a fast velocity at depth? Or does the sensitivity of the HSVR method stop at the top of the bedrock? I understand that you get a peak because of energy bouncing around in a low-velocity layer. …. So another alternative was to look at Figure 8 is a mask below bedrock"*
   **Response:** Yes, the inversion of HVSR curves do confirm fast velocity at depth. While the sensitivity of the HVSR curve diminished after the bedrock, the modelling of the half-space during the inversion scheme still reveal a consistent Vs value in the halfspace which also correlates with high bedrock velocity. It is highly unlikely that the interpreted dikes are a lateral homogeneity such as Tor as we observe no such feature during the field survey. Sensitivity of the HVSR method does diminished when we reach a homogenous lithology such as a bedrock. However, the inverted HVSR curves does determined the bedrock velocity which is of our interest. However, the bedrock (or mathematically can be referred to as the half-space) velocity modelled from the inversion routine can still be considered reliable even until after several meter below the top of the bedrock. Additionally, as per the second

reviewer comments, we now exclude the modelled velocity with high uncertainty. (Line 210 – 213, Figure 6)

10. **Comments:** *L335 "I am not sure how this is supported; you say tectonic stress or lithostatic decompression, but the way the manuscript is currently written, it's tough to know where and why the authors arrived at this conclusion.*

    *L338 "However, our study site in Pan de Azúcar provides an excellent observation of conditions preceding chemical weathering" How do you know? Is this relying on the cross-cutting relationships of the mafic dikes? Is it because of the timing of magma emplacement? It's really not clear to me why this is a good example and how you even know those are tectonic fractures versus weathering fractures. Does climate come into play? This is what's said on L340, "These, however, were absent in Pan de Azúcar given the absence of water and vegetation". This needs to be moved up in the paragraph—the authors should lead with this."*

    **Response:** We agree that this part of discussion can be improved to help the reader understand the thought process that leads to the hypothesis. Additionally, we also support this paragraph by adding the CDF data which shows no weathering process comes into play in our study area. (Section 6.6, Figure 4, Figure 5D)

11. **Comments:** *L334 – 359 "The authors are pulling from a lot of observations elsewhere that are not shown in this paper, so they need to describe some specifics from those cases to help hit this point home. In other words, please include relevant information so that this paper can stand alone.."*
    **Response:** We agree that a more explicit comparison between the different sites should be included to this manuscript. We added a figure which provides a climate and weathering structure comparison between the different sites. (Figure 10, Section 6.6)

**Reviewer 2 (Anonymous)**

We thanked the first reviewer for the extensive review which helps in improving our manuscript. We provided the following response to the comments:

**General Comments:**

1. ***Comments:*** *"However, it is somewhat disappointing that the presented results do not include resolution test (e.g., checkerboard test. See Wang et al., 2021) "*
**Response:** We accept this comment and have now include an uncertainty analysis of the model. Unfortunately, resolution test similar to the one presented in Wang et al. (2021) is not possible as we used a single station surface wave method which does not constitute the use of ray based resolution test such as checkerboard test. Instead, we have now included the model uncertainty analysis to limit our resolved models. (Figure 6 – 9)

2. ***Comments:*** **"** *It would be great to compare shear wave velocity profile in the borehole with the 1D profiles extracted at nearby stations. If any information available, please add that comparison into the manuscript. "*
**Response:** Unfortunately, we do not have a sonic log information from the borehole.

**Line Specific Comments:**

1. ***Comments:*** *L24 " Add "shear wave velocity (Vs)"."*
**Response:** We revised accordingly. (Line 19)

2. ***Comments:*** *L44 – 45 "Not clear what authors wanted to report here."*
**Response:** We revised the sentences accordingly to make it more clear. (Line 44)

3. ***Comments:*** *L50 – 53 "Not clear. Re-write the sentence."*
**Response:** We rephrase the sentences to make it more clear for the reader. (Line 50- 54)

4. ***Comments:*** *L89 – 90 "Use subscript (e.g., Na2O)"*
**Response:** We revised accordingly. (Line 89 - 90)

5. ***Comments:*** *L104 – 106 "Not clear what authors wanted to report here."*
**Response:** We revised the sentences accordingly to make it clearer. (Line 103 - 104)

6. ***Comments:*** *L203 "Remove "quickly""*
**Response:** We revised accordingly.

7. ***Comments:*** *L233 "Define "masl"."*
**Response:** We added the explanation. (Line 278)

8. ***Comments:*** *L116 "Define "SESAME""*
**Response:** We added the definition. (Line 116)

9. ***Comments:*** *L238 "Please add velocity contour lines into figure 7b to show 3 layers discussed in the text."*

**Response:** We added the contour line to the figure. (Figure 8 and 9)

10. ***Comments:*** *L307 "Replace "also ask" with "assume". I noted similar types of wording at many places in the manuscript (e.g., sentence start with "We found…" in line 377) and I would recommend fixing those."*
    **Response:** We revised the phrases accordingly across the manuscript.

11. ***Comments:*** *L293 "What is "ϑρο"."*
    **Response:** This is a typo and we have now removed it.

**Figure Comments**

12. ***Comments:*** *Fig 1: Caption- Replace "Red dots" with "Black dots".*
    **Response:** We revised the caption accordingly to make it more clear.

13. ***Comments:*** *Fig 4: Add shading (e.g., gray color) to the area where you have less data coverage (white region in your plots) or lower resolution (see example figures in Wang et al., 2021).*
    **Response:** Thank you for the suggestion. We revised the figure accordingly and gray-shaded the area that has no data coverage or has an uncertainly > 0.3 km/s. More details are discussed in the text.

14. ***Comments*** *Fig 5: Add data point location into figure 1. If Vs profile beneath the borehole is available, pleas add that into figure B for comparison.*
    **Response:** We revised the figure as suggested. Unfortunately, we do not have sonic logs available in the borehole.

15. ***Comments*** *Fig 6: Add shading as I mentioned above for Fig 4. Please add color scales corresponding to each slice. If you use same color scale for all figures, then, put only one-color scale bottom of the figure. Fig A-C needs x-axis labels. Remove words top on each figure (e.g., Depth Slice DS: XX) and add depth value upper right corner of each figure as "10 m or 20 m etc." (see example figures in Wang et al., 2021). Keep black solid line (cross section) within the area where you have the best resolution*
    **Response:** Thank you for the suggestion. We revised the figure accordingly. However, we maintain the cross section as it is as we wanted to present the cross section that cuts through the borehole location and is perpendicular to the possible dike structure.

16. ***Comments*** *Fig 7 & 8: For Fig A; similar comment as Fig 6.*
    *For Fig B; Zoom into the region where you have the best resolution. It seems like X axis is not correct. Remove text above the figures B and C. Mark 'NW' and 'SE' corner of the profile (see example figures in Wang et al., 2021). Remove depth axis in Fig B and C.*
    **Response:** We revised the figures as suggested. However, we maintain the depth axis as we used it as reference in the discussion part. We also maintain the selected cross section as we want to show the possible dike structure that is more pronounced in this cross section.